# Understanding the needs of women undergoing breast ultrasound: Are male radiologists still needed?

Elisabeth Sartoretti[1,2⊙], Selina Largiadèr[1⊙], Thomas Sartoretti[1,2], Simin Laures[3], Martin Alexander Walter[1,4], Eva Monti[1], Ina Füchsel[5], Mira Dettling[5], Stephan Pfister[4], Peter Dubsky[6], Alexander Ort[1], Sabine Sartoretti-Schefer[5], Matthias Meissnitzer[7], Klaus Hergan[7], Rosemarie Forstner[7], Simon Matoori[8], Rasmus Bech- Hohenberger[9], John M. Froehlich[4], Tino Plümecke[1], Dorothee Harder[10], Dow Mu Koh[11], Andreas Gutzeit[1,4,6,12] *

**1** Department of Health Sciences and Medicine, University of Lucerne, Lucerne, Switzerland, **2** Faculty of Medicine, University of Zurich, Zurich, Switzerland, **3** Institute of Radiology, Cantonal Hospital Aarau, Aarau, Switzerland, **4** Institute of Radiology and Nuclear Medicine, Hirslanden Klinik St. Anna, Lucerne, Switzerland, **5** Department of Radiology, Cantonal Hospital Winterthur, Winterthur, Switzerland, **6** St. Anna Breast Center, Hirslanden Klinik St. Anna, Lucerne, Switzerland, **7** Department of Radiology, Paracelsus Medical University Salzburg, Salzburg, Austria, **8** Faculté de Pharmacie, Université de Montréal, Montréal, Canada, **9** Faculty of Medicine, University of Copenhagen, Copenhagen, Denmark, **10** Department of Radiology, University Hospital of Basel, University of Basel, Basel, Switzerland, **11** Department of Radiology, Royal Marsden Hospital, Surrey, United Kingdom, **12** Department of Chemistry and Applied Biosciences, Institute of Pharmaceutical Sciences, ETH Zurich, Zurich, Switzerland

⊙ These authors contributed equally to this work.
* andreas.gutzeit@unilu.ch

**Data Availability Statement:** All relevant data are within the manuscript and its Supporting Information files.

## Abstract

### Introduction

A trend towards less male radiologists specializing in breast ultrasound was observed. A common notion in the field of breast radiology is, that female patients feel more comfortable being treated by female radiologists. The aim of the study was to understand and report the needs of women undergoing breast ultrasound with regards to the sex of the radiologist performing the investigation.

### Methods

Informed consent was obtained from all patients prior to inclusion in a prospective bi-center quality study. At center 1 (72 patients), the women were examined exclusively by female radiologists, at center 2 (100 patients) only by male radiologists. After the examination the patients were asked about their experiences and their wishes for the future.

### Results

Overall, women made no distinction between female and male radiologists; 25% of them wanted a female radiologist and 1.2% wanted a male radiologist. The majority (74%) stated that it made no difference whether a female or male radiologist performed the examination. The majority of women in group 2, who were investigated exclusively by male radiologists,

**Funding:** The authors received no specific funding for this work.

**Competing interests:** The authors have declared that no competing interests exist.

**Abbreviations:** BI- RADS, Breast Imaging Reporting and Data System.

stated that they had no preferences with regard to the sex of the radiologist (93%); 5% of the women wished to be investigated solely by a female radiologist and 2% exclusively by a male radiologist.

## Discussion

The majority of women undergoing breast ultrasound are unconcerned about the radiologist's sex. It would appear that women examined by male radiologists are less selective about the sex of the examining radiologist.

## Trial registration

Written informed consent was obtained from all patients. All patient data were anonymized. The physicians had no access to any further personal data. National regulations did not require dedicated ethics approval with anonymized lists or retrospective questionnaires.

## Introduction

Mammography is the gold standard for breast cancer screening and diagnosis in women [1]. However, an additional ultrasound screening may be recommended in specific situations, such as women with a higher breast density, high-risk populations, or ambiguous findings on mammography [2]. In recent years, it has become increasingly clear that more attention should be given to the welfare of female patients in medicine because they have different needs, physiological conditions, and respond differently to treatment than male patients [3–5]. These differences are addressed in the field of gender medicine [6].

Despite scarce evidence, female patients are consistently reported to dislike consulting male physicians and would be more likely to prefer a female physician [7–9]. This is also evident in surveys of gynecologists: while 7% of gynecologists in 1970 were female, currently women comprise about 60% of gynecologists [10].

Male radiologists also appear to be less interested in female imaging and breast care than their female counterparts, although we lack official data in this regard. However, an inquiry at the European Society of Radiology (ESR) regarding the sex ratio of their memberships and specialization revealed that the female/male ratio of the ESR, which currently counts about 123,500 members, is 39/61%. In contrast, the European Diploma in Breast Imaging (EDBI) had a majority of accreditations being given to women in recent years, with a female/male ratio of 70/30% [11].

The reasons for this are not clear. Apart from personal interest, the perceived demands and needs for specific radiological personnel have a significant influence on a resident's intentions to be trained in a sub-specialty. When asked about this, male doctors showed uncertainty as to whether they were still desired by patients in a field with a majority of female patients.

The aim of the present study was to understand and report the needs of women undergoing breast ultrasound with regard to the sex of the radiologist performing the investigation.

## Methods

### Study subjects

This prospective two-center study was conducted in accordance with the Declaration of Helsinki. Written informed consent was obtained from all participants. All participant data were

anonymized as part of a routine quality assurance measure at the institutions. According to national regulations, no explicit ethics approval is required under these conditions. The study was conducted at the hospital (Cantonal Hospital Winterthur): Center 1, and the breast center at hospital (Breast center Lucerne): Center 2. Centers 1 and 2 are certified breast imaging centers performing approximately 5000 and 9000 breast examinations, respectively, each year.

A total of 172 participants were included between August 2021 and May 2022. Inclusion criteria were the following: a) consecutive participants referred to the radiology department for mammography and ultrasound; b) participants consented to a complete structured telephone interview with a trained team of female quality managers 1 to 7 days after the examination; c) only participants with BI-RADS 1 or 2 were included. The reason was that participants from BI-RADS 3 onwards are subject to significant stress and the latter could influence the results of the study. Furthermore, their worry might limit their willingness to be interviewed. Exclusion criteria were the following: a) participants younger than 18 years of age; b) participants unwilling to speak to the quality mangers in a telephone interview; c) participants with BI-RADS 3 lesions or worse on ultrasound; d) participants not available for an interview during the mentioned period.

## Study design

Prior to the examination, participants were asked if they were willing to participate in a telephone interview following their examination.

The initial mammography was performed in centers 1 and 2 on a Philips Mammomat Inspiration or a SenoClaire® digital breast tomosynthesis system of Siemens Healthcare. The mammography was evaluated immediately by the radiologist A.G, S.P.,I.F., M.D., and was followed by a standard ultrasound examination performed on a Voluson E8 machine, General Electric Healthcare, or a GE Logiq E9. The participants were not personally acquainted with the radiologists.

After application of the exclusion criteria (Fig 1), 172 participants were included in the study (72 at center 1 and 100 at center 2).

## Center 1: Participants investigated exclusively by female radiologists

At this center the female participants were cared for exclusively by female radiologists and technicians. Each of the two female board-certified radiologists at this center I.F., M.D had 14 years of experience in specialized ultrasound investigations. The doctor-participant conversation was conducted by each radiologist without prior standardization or agreement. Mammography was performed by a female technician. The numbers of ultrasound examinations were divided equally between the two female radiologists. Participants were not free to choose a radiologist. Unknown to the participant, the time taken for the ultrasound investigation was measured with a stopwatch. The experimental setup is illustrated in Fig 2.

## Center 2: Participants investigated exclusively by male radiologists

The ultrasound investigations were performed exclusively by male radiologists. The two male certified male radiologists A.G.,S.P. had 25 and 20 years of experience, respectively, in specialized breast ultrasound. Before starting the examination, the female participants were asked whether they were willing to be examined by male radiologists. No participant refused or asked for a female radiologist. The doctor-participant communication during the procedure was not standardized. The mammography was performed by a female technician. The numbers of ultrasound examinations were divided equally between the two male radiologists.

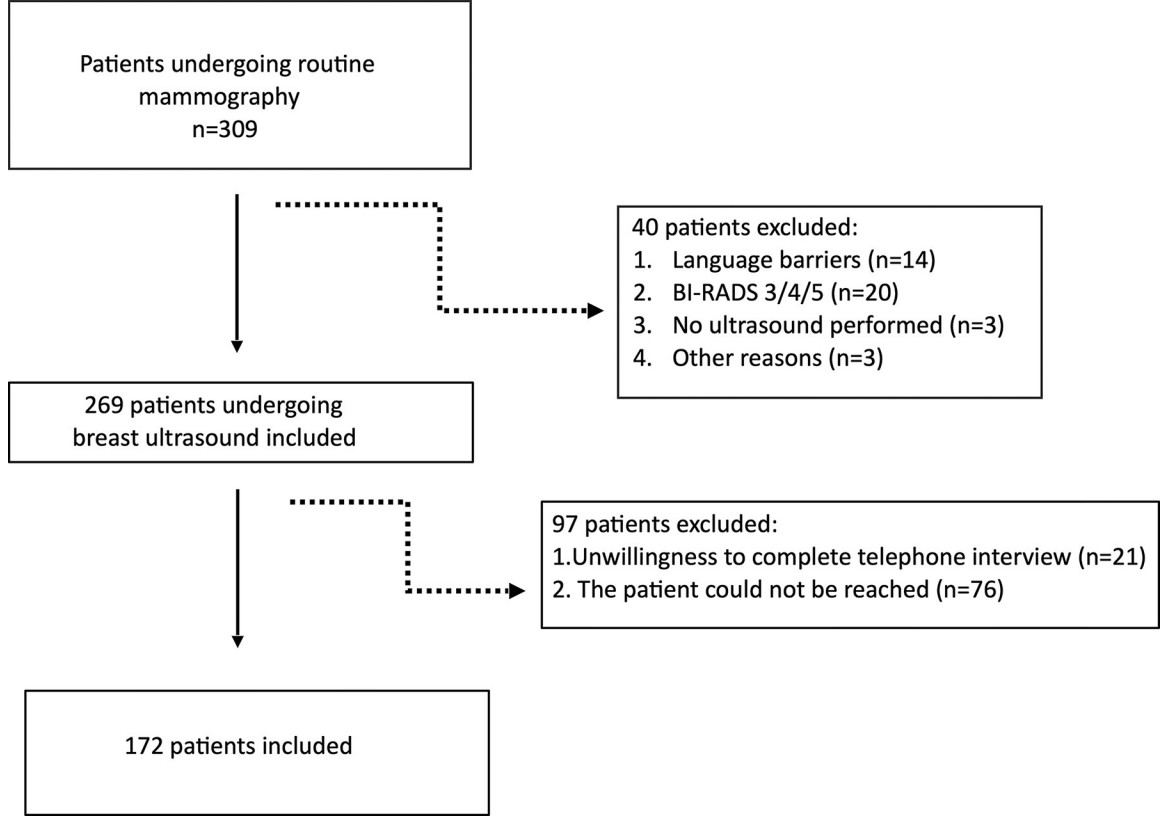

**Fig 1. Flow diagram showing the inclusion and exclusion criteria.** This diagram depicts the inclusion and exclusion criteria of the included participants.

Unknown to the participant, the time taken for the ultrasound investigation was measured with a stopwatch. The experimental setup is shown in Fig 3.

## Telephone interview

At 1–7 days after the imaging examination, each participant received a telephone call. The interviews were conducted by three trained female interviewers from the quality management staff at the institution. A total of 5 questions were addressed to each patient and the responses recorded. The duration of the interview, and the duration of the interview relative to the imaging studies were documented. The questions and responses are summarized in Table 1.

The questions (Table 1) were developed by a board-certified senologist with a master's degree in psychology and three years of experience in psychology and data management A.G.

## Statistical analysis

Data distribution was checked visually by means of boxplots, histograms and barplots. To check for differences in categorical, polytomous data between the two centers, two-tailed Fisher's exact tests were computed. To check for differences in continuous, ordinal data between the two centers, two-tailed Mann-Whitney-U-Tests were computed. P-values $<0.05$ were considered significant. All analyses were performed with the statistical software R (version 3.3.3) (R Core Team, 2017).

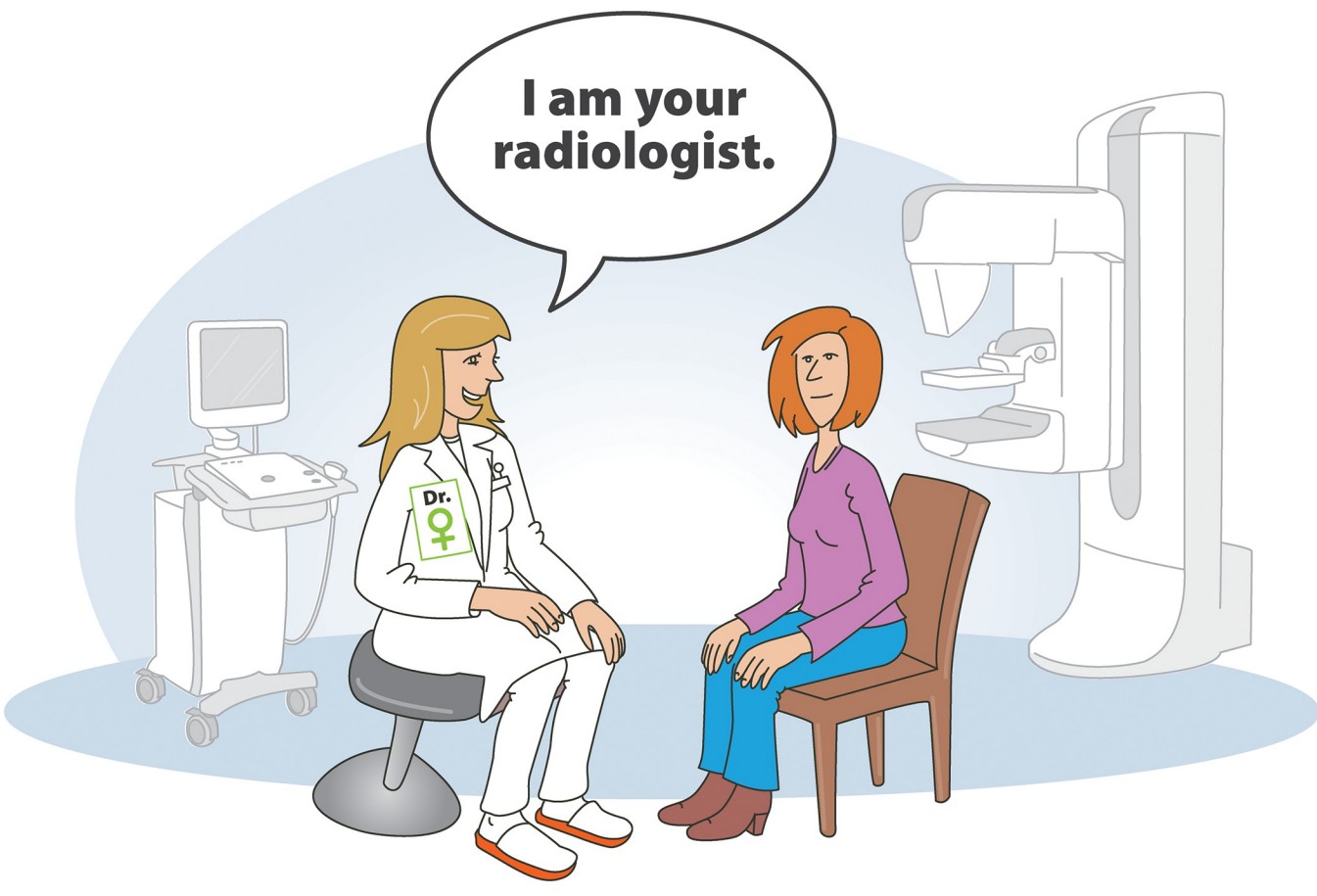

**Fig 2. Study set up in Center 1.** In Center 1 the participants were cared for exclusively by female radiologists. The doctor- participant communication was done without prior standardization between radiologists. Patients had no choice with regards to which radiologist undertook their examination as part of the normal routine.

## Results

Inclusion and exclusion criteria are shown in Fig 1.

After exclusion a total of 172 participants remained. 72 participants in center 1 (mean age: 61.3 ±11.9 years), 100 participants in center 2 (mean age 60.8 ± 10.4 years) remained.

The mean duration of the ultrasound examination and the discussion of its findings was 528.2 (9 minutes) ± 107.6 sec. (range: 240–960 sec.) The duration of discussion did not differ between center 1 and 2 (p > 0.05). The telephone interviews were conducted on average 377.5 ± 254.7 sec. (6 minutes.) (Range: 155–2100 sec.). The questions and responses are summarized in Table 1.

Women who undergo breast ultrasound do not care whether a female or a male radiologist performs the examination (female radiologist only 25%, male radiologist only 1.2%, does not matter 74%). However, the follow-up inquiry of women who were exclusively attended to by female radiologists at center 1 suggests that women would still prefer to be cared for by women (52% only female radiologist, 0% only male radiologist, 47% does not matter). On the other hand, at the follow-up inquiry of women who underwent breast ultrasound examinations performed exclusively by a team of male radiologists (center 2), the majority of the women had no specific preferences about the sex of radiologist at future visits and did not wish to be served exclusively by a female or a male radiologist in the future (female radiologist only 5%, male radiologist only 2%, does not matter 93%).

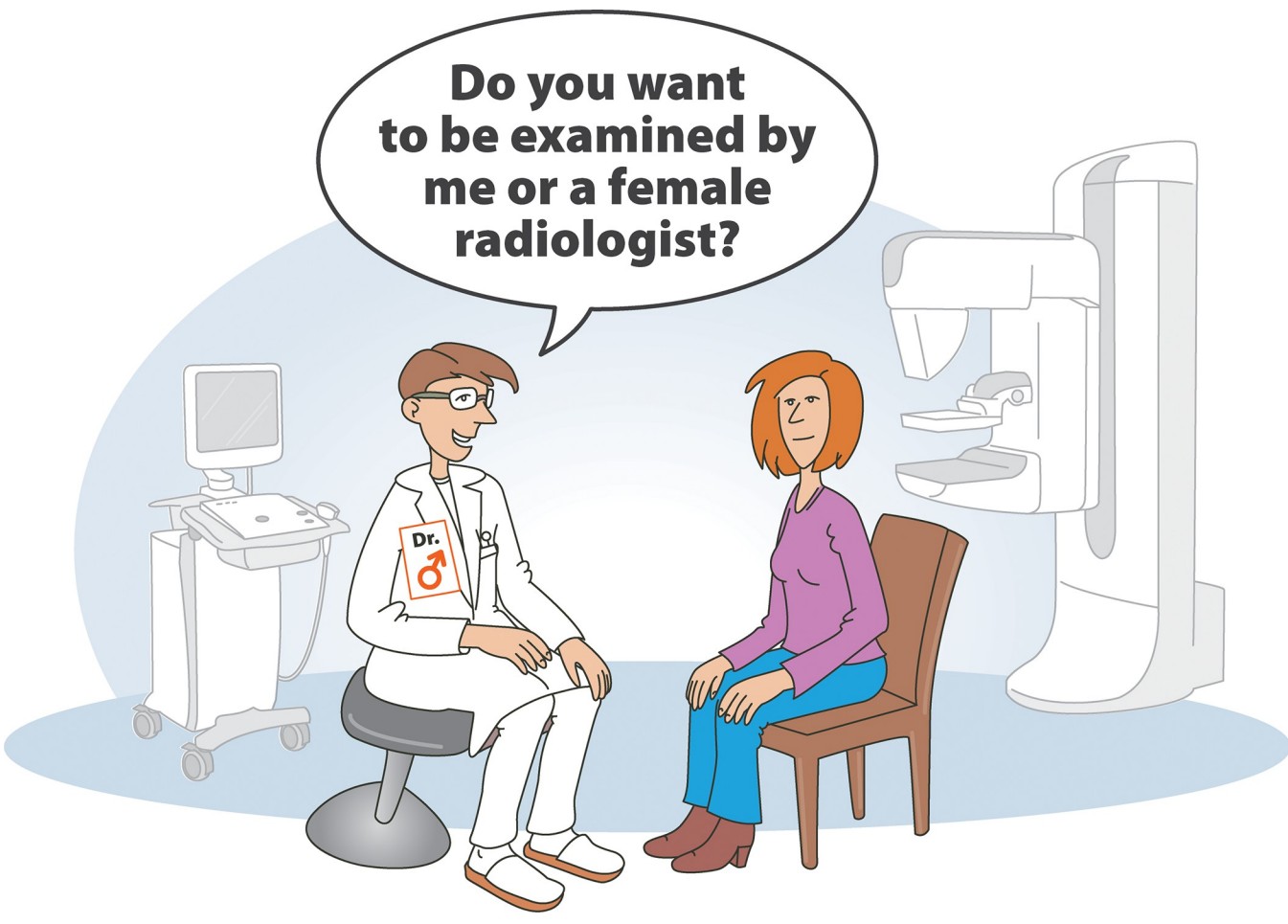

**Fig 3. Study set up in Center 2.** In Center 2, female participants were managed exclusively by male radiologists. Before starting with ultrasound examination, participants were asked, if it was okay for them as female patients to be examined by a male radiologist. No woman refused to be examined by a male radiologist in our study, although there would have always been an alternate female radiologist available.

Our results showed that a minority of women would like to undergo breast examinations exclusively by a female or a male radiologist. Therefore, male radiologists routinely ask the patient whether she finds the setting acceptable. Considering the data reported from center 2 in our study (2% wished to see a male radiologist), it would be appropriate for a female radiologist to ask the patient the same question. The scenario is depicted in Fig 4 as a suggestion.

Many participants said that they did not care about the doctor's sex as long as he/she was friendly and competent. In both groups, the final discussion with the radiologist was extremely important to the women, regardless of whether the radiologist was male or female (p = 0.9).

The distribution of female and male gynecologists among the surveyed participants was relatively balanced. However, slightly more participants at center 1 preferred a female gynecologist, while the views were more balanced at center 2. At center 2, the participants were indifferent to the sex of the gynecologist. The impact of these data on radiological care cannot be evaluated at the present time.

At both centers, a clear majority preferred to undergo mammography by a female technician. It should be noted that, in Switzerland, this investigation has been the domain of female technicians for many years. Male technicians, despite their potential interests or skills, in most cases are not permitted to perform this investigation.

**Table 1. Summary of the telephone interviews.**

| | All patients (n = 172) | Center 1 Female Radiologists (n = 72) | Center 2 Male Radiologists (n = 100) | Fischers exact test for Center 1 versus Center 2 |
|---|---|---|---|---|
| What is the sex of your gynecologist/ general physician? (male/female/other) | female: 112 (65.1%) male: 60 (34.9%) other: 0 (0%) | female: 53 (73.6%) male: 19 (26.4%) other; 0 (0%) | female: 59 (59%) male: 41 (41%) other: 0 (0%) | P = 0.053 |
| Did you select a gynecologist or general physician because of his/her sex (man/woman) or could you also imagine being cared for by the opposite sex? | only female: 64 (37.2%) only male: 10 (5.8%) I do not care: 98 (57%) | only female: 32 (44.4%) only male: 8 (11.2%) I do not care: 32 (44.4%) | only female: 32 (32%) only male: 2 (2%) I do not care: 66 (66%) | P = 0.003 |
| The mammography was performed by a female technician. What would you prefer in the future? (only female technician, only male technician, I don't care)? | only female: 108 (62.8%) only male: 0 (0%) I do not care: 64 (37.2%) | only female: 50 (69.4%) only male: 0 (0%) I do not care: 22 (30.6%) | only female: 58 (58%) only male: 0 (0%) I do not care: 42 (42%) | p = 0.15 |
| Who would you like to have as your attending radiologist in the future? (only female radiologist, only male radiologist, I don't care) | only female: 43 (25%) only male: 2 (1.2%) I do not care: 127 (73.8%) | only female: 38 (52.8%) only male: 0 (0%) I do not care: 34 (47.2%) | only female: 5 (5%) only male: 2 (2%) I do not care: 93 (93%) | p<0.001 |
| How important was the discussion with the radiologist? (1 = unimportant, 5 = very important) | 5; (5.5)– 4.84 ± 0.48 | 5; (5.5)– 4.83 ± 0.5 | 5; (5.5)– 4.84 ± 0.47 | P = 0.94 |

This table summarizes the interview questions and answers given. The results are divided into the following categories: all patients, Center 1 and Center 2.

## Discussion

Women undergoing an ultrasound investigation of the breast do not have a specific preference for a male or female radiologist. Overall, 73% percent of women did not care whether the radiologist is female or male. Additionally at center 2, which was manned exclusively by male radiologists, the majority of the participants at this center did not care about the sex of the radiologist in future investigations (93%, p< 0.05).

Patients who undergo an investigation of the breast are almost exclusively women, but occasionally also men. To our knowledge, no previous study has addressed the sex-specific needs of women undergoing radiological or ultrasound investigations of the breast. A few studies in gynecology have addressed this topic. Although we lack robust data, it is widely assumed that female patients reject male physicians. It has been reported that up to 93% of women wish to be treated by a female physician and reject the notion of a male physician [8, 9, 12–14]. However, this notion was not confirmed in the present study. At our center 2 about 66% of the women were indifferent to the sex of the gynecologist. In other words, the above-mentioned studies may not reflect the preferences of our study cohort. Notably, many of the cited study groups examined women from not Western countries and the authors were largely women. While the authors have no intention of expressing a cultural or sex-specific bias, it should be noted that a society focused on the exclusive care of women by women may influence the future expectations of patients. This was also observed at our center 1, where only women were available to perform the examination. By contrast, the women at center 2, which was examined by male radiologists, had more diverse opinions. By contrast, the women at

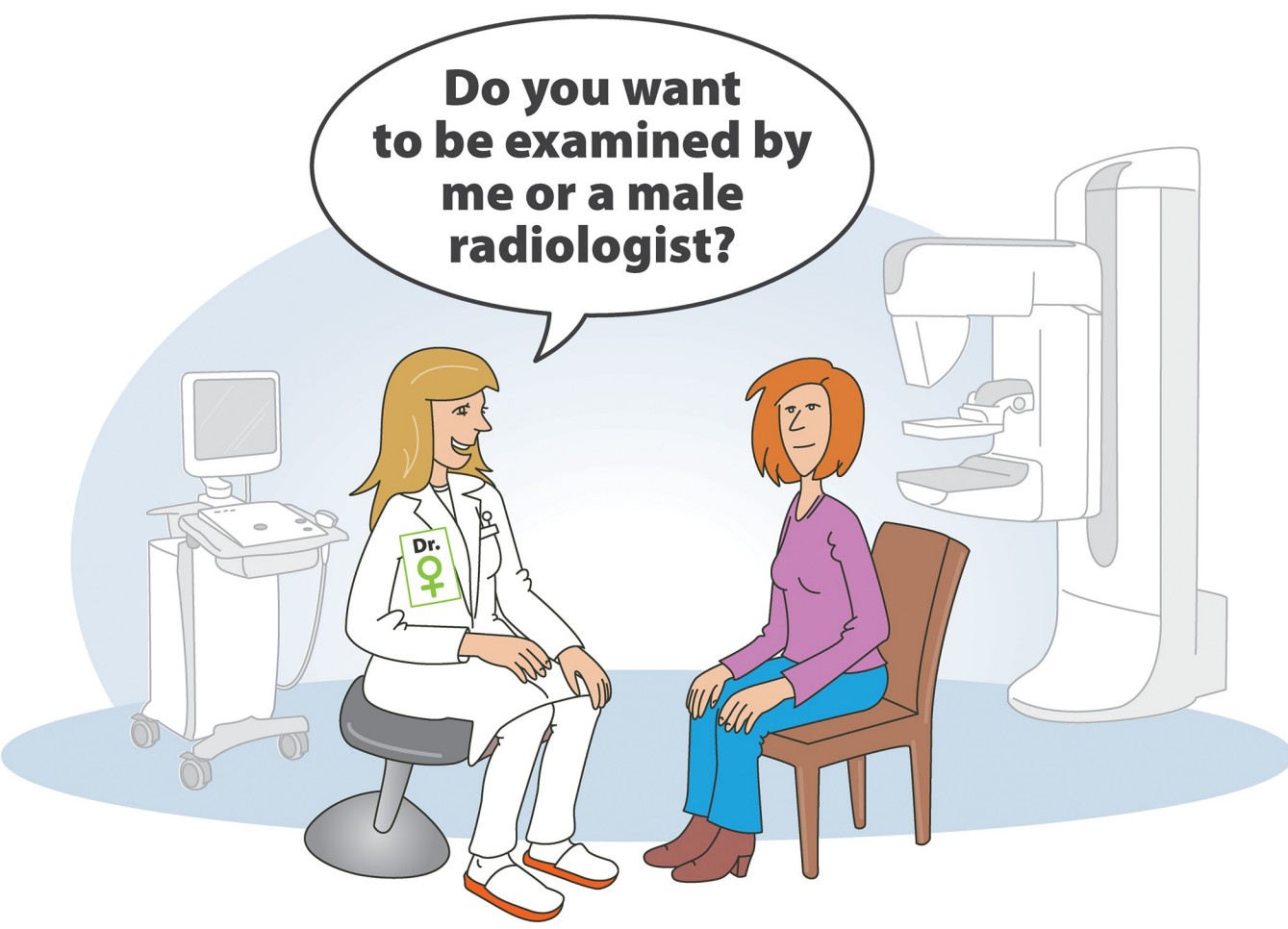

**Fig 4. Ideal study set up for all centers.** In Center 2 a small proportion of patients indicated that they would like to be examined exclusively by a male radiologist. To respect the needs of all patients, female radiologists should also ask about patient's preference concerning the sex of their radiologist.

center 2, which was serviced by male radiologists, had more diverse opinions concerning the sex of their attending radiologist.

An important reason to have male radiologists in ultrasound teams is that male patients are also investigated in radiology departments; this is due to the increasing incidence of male breast cancer throughout the world [15–18]. Furthermore, increasing numbers of transsexual women with hormonal stimulation are likely to develop breast cancer [19–21]. The prevalence of gynecomastia among men is also high, and these men have to undergo ultrasound investigations [22]. For this reason, a breast cancer screening program for men and transgender women is already being discussed. Given the fact that mixed patients cohorts are examined by specialized ultrasound teams, it would be appropriate to have male doctors in radiology teams. This is consistent with the need for health care systems worldwide to address sex-related inequalities and restrictive sexual norms in the medical profession. The phenomenon affects female and male doctors to the same degree [23].

Based on the presented data, we assume that the discussion about the radiologist's sex is not of that great importance to female patients. According to the feedback from our patients at both centers, the competence expressed by the radiologist is much more important than his/her sex. Few male radiologists are interested in breast ultrasound. This might be due to the

notion that female patients do not give preference to male doctors. However, the data obtained in the present investigation show that male specialists are quite welcome.

The present investigation permits no conclusive statement as to whether these data will have an impact on women's preferences for a female or male gynecologist, or whether they will have no preferences in this regard. In fact, while 37% of patients wish to be cared for by a female gynecologist, only 6% wished for a male gynecologist. But the majority of women appear to be indifferent to the sex of the gynecologist (57%). With regard to radiographers, the majority (63%) of women at both centers would like to be examined by a female technician. Again, the body of robust data in this regard is sparse. However, the few published data are similar to ours. In one study, 46% of women said they would be uncomfortable if a man performed the mammography [24]. However, in a large Australian study, male radiographers were used to perform mammography when there was a shortage of radiographers, and the majority of female patients had no problems with a male radiographer [25]. As with radiologists in center 2, the acceptance level of female patients increases when they come into contact with mixed, competent teams and have good experiences.

The limitations of this study are, that the investigation was limited to two centers. We assume that the results significantly depend on the social structure. Therefore, it would be appropriate to perform further surveys in other countries and determine the respective regional needs of female patients. Additionally, patients at centers 1 and 2 were only cared for by two radiologists each, who were highly experienced. More radiologists with different degrees of training should be included in future investigations.

## Conclusion

We believe that the notion of female patients giving preference to female radiologists for their breast ultrasound examination is incorrect and does not take the actual needs of female patients into account. The majority of female patients were indifferent to the sex of the radiologist and were much more willing to accept the doctor's sex when they were examined by mixed teams. Female patients are much more willing to accept the doctor's sex when they are examined by mixed teams. We advocate mutual tolerance and efforts to train more male specialists in this fascinating and increasingly important subspecialty of radiology. From the patient's point of view, a competent radiologist is most welcome regardless of his/her sex.

## Supporting information

**S1 Checklist. *PLOS ONE* clinical studies checklist.**
(DOCX)

## Author Contributions

**Conceptualization:** Elisabeth Sartoretti, Selina Largiadèr, Thomas Sartoretti, Simin Laures, Klaus Hergan, John M. Froehlich, Dorothee Harder, Andreas Gutzeit.

**Data curation:** Selina Largiadèr, Thomas Sartoretti, Klaus Hergan, Tino Plümecke, Dow Mu Koh, Andreas Gutzeit.

**Formal analysis:** Alexander Ort, Klaus Hergan, Dow Mu Koh, Andreas Gutzeit.

**Investigation:** John M. Froehlich, Andreas Gutzeit.

**Methodology:** John M. Froehlich, Andreas Gutzeit.

**Project administration:** Andreas Gutzeit.

**Resources:** Andreas Gutzeit.

**Software:** Andreas Gutzeit.

**Supervision:** Rasmus Bech- Hohenberger, John M. Froehlich, Andreas Gutzeit.

**Validation:** Rasmus Bech- Hohenberger, Andreas Gutzeit.

**Visualization:** Rasmus Bech- Hohenberger, Andreas Gutzeit.

**Writing – original draft:** Elisabeth Sartoretti, Selina Largiadèr, Thomas Sartoretti, Simin Laures, Martin Alexander Walter, Eva Monti, Ina Füchsel, Mira Dettling, Stephan Pfister, Peter Dubsky, Alexander Ort, Sabine Sartoretti-Schefer, Matthias Meissnitzer, Rosemarie Forstner, Simon Matoori, Rasmus Bech- Hohenberger, Tino Plümecke, Dorothee Harder, Dow Mu Koh, Andreas Gutzeit.

**Writing – review & editing:** Thomas Sartoretti, Tino Plümecke, Dorothee Harder, Andreas Gutzeit.

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
