## [Decision Letter · Decision Letter 0]

29 Jun 2023

PONE-D-23-14827Understanding the needs of women undergoing breast ultrasound: Are male radiologists still needed?PLOS ONE

Dear Dr. Gutzeit,

Thank you for submitting your manuscript to PLOS ONE. After careful consideration, we feel that it has merit but does not fully meet PLOS ONE’s publication criteria as it currently stands. Therefore, we invite you to submit a revised version of the manuscript that addresses the points raised during the review process.

We look forward to receiving your revised manuscript.

Kind regards,

Aloysius Gonzaga Mubuuke

Academic Editor

PLOS ONE

Journal Requirements:

"We have no fundings to declare. " 

"No authors have competing interests. "

5. Please amend the manuscript submission data (via Edit Submission) to include author Dr. Froehlich John M.

6. We note that Figures 2,3 and 4 in your submission contain copyrighted images. All PLOS content is published under the Creative Commons Attribution License (CC BY 4.0), which means that the manuscript, images, and Supporting Information files will be freely available online, and any third party is permitted to access, download, copy, distribute, and use these materials in any way, even commercially, with proper attribution. For more information, see our copyright guidelines: http://journals.plos.org/plosone/s/licenses-and-copyright.

a. You may seek permission from the original copyright holder of Figures 2,3 and 4 to publish the content specifically under the CC BY 4.0 license. 

Additional Editor Comments:

The reviewers have provided useful comments to improve the paper. In addition to these, please explain clearly the statistical methods that were used to the reader.

Reviewers' comments:

Reviewer's Responses to Questions

**Comments to the Author**

1. Is the manuscript technically sound, and do the data support the conclusions?

Reviewer #1: Yes

Reviewer #2: Yes

2. Has the statistical analysis been performed appropriately and rigorously? 

Reviewer #1: I Don't Know

Reviewer #2: Yes

3. Have the authors made all data underlying the findings in their manuscript fully available?

Reviewer #1: Yes

Reviewer #2: Yes

4. Is the manuscript presented in an intelligible fashion and written in standard English?

Reviewer #1: Yes

Reviewer #2: Yes

5. Review Comments to the Author

Reviewer #1: Thanks for this interesting study on understanding the needs of women undergoing breast ultrasound. The study included consecutive participants referred to the radiology department for mammography and ultrasound at two certified breast imaging centers performing approximately 5000 and 9000 breast examinations each year between August 2021 and May 2022. Were there any male participants or transsexual women seen?

The authors blinded the country where this study was done but it would be good to provide data on how many males/ transsexual women are referred for breast imaging to provide context for this study and the recommendations made. Especially since the authors advocate for efforts to train more male specialists in this field (lines 343 and 344).

The authors state that after application of the exclusion criteria (Figure 1), 172 participants were included, however, the quoted figure 1 shows 174 participants were included.

The statistical analysis section of the write up is not very informative. More detail on exactly how data was handled would be helpful.

It seems as though the women prefer a female physician where the option is available. For example; majority of the women in both centers 1 (69.4%) and 2 (58%) preferred a female technician to do their mammogram arguably because a female technician was available. Although a small percentage 30.6% in center 1 and 42% in center did not care about the sex of the practitioner, no woman opted for a male technician.

Majority of the women (52.8%) in center 1 where female radiologists were available still preferred only female compared to only 2% who preferred male in center 2 where the male radiologists were available.

Perhaps the women 'don't care' simply because they know their preferred option may not be available.

Please provide reference for this statement "Based on the published literature, we assume that the discussion about the radiologist’s sex is entirely irrelevant to female patients" in lines 311 and 312. As well as this "Few male radiologists are interested in breast ultrasound" (line 314).

Repetition in lines 296- 298.

Reviewer #2: I am pleased to recommend accepting the submitted article for publication. The study presents the results of original research, providing valuable insights into the topic. The findings reported in this study have not been published elsewhere, which adds to the existing body of knowledge. My only concern is the large number of authors, I would like to know what their contributions were.

The experiments, statistics, and other analyses conducted in this research are of good technical standard. The authors have described these methods with sufficient detail, allowing for reproducibility and understanding of the study design. The data presented in the article support the conclusions drawn by the authors in an appropriate and logical manner.

One notable strength of this article is its clear and intelligible presentation. The authors have effectively communicated their research in standard English, making it accessible to a wide range of readers. The article also adheres to the ethical standards of experimentation and research integrity, demonstrating the authors' commitment to conducting their study in an ethically responsible manner.

In summary, this article fulfills the essential criteria for acceptance. It presents original research with good findings, good technical standards, appropriate conclusions supported by data, clear and accessible language, adherence to ethical standards, and compliance with reporting guidelines. Therefore, I recommend accepting this article for publication.

6. PLOS authors have the option to publish the peer review history of their article (what does this mean?). If published, this will include your full peer review and any attached files.

Reviewer #1: No

Reviewer #2: **Yes: **Elsie Kiguli-Malwadde

---

## [Author Response · Author response to Decision Letter 0]

27 Jul 2023

Dear ladies and gentlemen

We gave detailed comments in the section: "answer to the editor".

Thank you very much.

---

## [Decision Letter · Decision Letter 1]

21 Aug 2023

Understanding the needs of women undergoing breast ultrasound: Are male radiologists still needed?

PONE-D-23-14827R1

Dear Dr. Gutzeit,

We’re pleased to inform you that your manuscript has been judged scientifically suitable for publication and will be formally accepted for publication once it meets all outstanding technical requirements.

Kind regards,

Aloysius Gonzaga Mubuuke

Academic Editor

PLOS ONE

Additional Editor Comments (optional):

Reviewers' comments:

Reviewer's Responses to Questions

**Comments to the Author**

1. If the authors have adequately addressed your comments raised in a previous round of review and you feel that this manuscript is now acceptable for publication, you may indicate that here to bypass the “Comments to the Author” section, enter your conflict of interest statement in the “Confidential to Editor” section, and submit your "Accept" recommendation.

Reviewer #1: All comments have been addressed

2. Is the manuscript technically sound, and do the data support the conclusions?

Reviewer #1: Yes

3. Has the statistical analysis been performed appropriately and rigorously? 

Reviewer #1: Yes

4. Have the authors made all data underlying the findings in their manuscript fully available?

Reviewer #1: Yes

5. Is the manuscript presented in an intelligible fashion and written in standard English?

Reviewer #1: Yes

6. Review Comments to the Author

Reviewer #1: I have no additional comments.

7. PLOS authors have the option to publish the peer review history of their article (what does this mean?). If published, this will include your full peer review and any attached files.

Reviewer #1: No

---

## [Editor Report · Acceptance letter]

17 Oct 2023

PONE-D-23-14827R1 

Understanding the needs of women undergoing breast ultrasound: Are male radiologists still needed? 

Dear Dr. Gutzeit:

I'm pleased to inform you that your manuscript has been deemed suitable for publication in PLOS ONE. Congratulations! Your manuscript is now with our production department. 

Kind regards, 

on behalf of

Dr. Aloysius Gonzaga Mubuuke 

Academic Editor

PLOS ONE